# Fragility Analysis Based on Damaged Bridges during the 2021 Flood in Germany

**Alessandro Pucci** [1,*], **Daniel Eickmeier** [2], **Hélder S. Sousa** [1], **Linda Giresini** [3], **José C. Matos** [1] **and Ralph Holst** [2]

1 Department of Civil Engineering, ARISE, ISISE, University of Minho, 4800-058 Guimarães, Portugal; jmatos@civil.uminho.pt (J.C.M.)
2 Federal Highway Research Institute (BASt), 51427 Bergisch Gladbach, Germany
3 Department of Structural and Geotechnical Engineering, Sapienza University of Rome, 00185 Rome, Italy
* Correspondence: ale@civil.uminho.pt

**Featured Application: The existing literature on bridge fragility curves for floods mainly uses analytical approaches. However, it is crucial to validate these models and to identify failure trends and patterns to detect vulnerabilities. Therefore, fragility curves obtained using data from actual collapses can be employed in CAT (catastrophe) models. Indeed, a gateway to faster recovery from bridge failures can be achieved by transferring the financial risk to insurance providers. Fragility curves allow the association of the hazard intensity to several damage levels, thus enabling the use of damage–loss equations.**

**Abstract:** Floods trigger the majority of expenses caused by natural disasters and are also responsible for more than half of bridge collapses. In this study, empirical fragility curves were generated by referring to actual failures that occurred in the 2021 flood in Germany. To achieve this, a calibrated hydraulic model of the event was used. Data were collected through surveys, damage reports and condition ratings from bridge owners. The database comprises 250 bridges. The analysis revealed recurrent failure mechanisms belonging to two main categories: those induced by scour and those caused by hydraulic forcing. The severity of the damage was primarily dependent on the bridge typology and, subsequently, on the deck's weight. The analysis allowed us to draw conclusions regarding the robustness of certain bridge typologies compared to others for a given failure mechanism. The likelihood of occurrence of the triggering mechanism was also highlighted as a factor to consider alongside the damage probability. This study sheds light on existing vulnerabilities of bridges to river floods, discussing specific areas in which literature data are contradictory. The paper also strengthens the call for a shift towards a probabilistic approach for estimating hydraulic force in bridge design and assessment.

**Keywords:** fragility curve; flood; scour; hydraulic force; bridge

## 1. Introduction

Several studies have reported floods as the main causes of bridge failures worldwide [1–3]. In addition, human-induced catchment modifications masked climate change, resulting in further difficulties predicting flood scenarios [4–6]. Although catastrophic floods often trigger risk-aversion behaviors by implementing virtuous management strategies [7], the same comes at a high societal and economical cost [8]. It is estimated that more than 120 million people worldwide are affected by floods each year, making it the most threatening natural hazard [9]. The vulnerability of society to floods highly depends on coping capacity, which is tightly linked to wealth indicators. Worldwide flood vulnerability research highlights that at the present climate change rate, inequalities among low and high-income countries will increase [10]. Concerning the European context, data from Risklayer

CATDAT reported that climate-related events are responsible for 80% of economic losses among those caused by natural hazards in Europe [11]. The highest economic damage per square kilometer occurred in Switzerland (~400 k EUR/km$^2$) and Germany and Italy (~300 k EUR/km$^2$). Of these, Germany and Switzerland had 37% of losses covered by insurance companies, while Italy had only 6%. Delegating the economic risk to insurance companies can be cost-effective for a faster recovery [12]. It is therefore of paramount importance to assess the economic risk of infrastructure facing extreme events [13]. Of these assets, bridges represent one of the most vulnerable elements [14]. Recently, failure scenarios were studied and systematized for small bridges in case of extreme events such as floods [15]. However, the performance of bridges against varying flooding scenarios is usually unknown, as the design is typically carried out on a deterministic basis [16]. On the contrary, fragility curves link the expected damage to a range of hazard intensities [17]. Subsequently, thanks to damage–loss equations, the financial aspect can be associated with the hazard intensity [18]. Therefore, fragility curves represent a milestone to financial risk assessment of structures [19–21]. The existing literature covers flood fragility curves for bridges mainly through analytical approaches and often in combination with earthquake hazards [22–24]. Nevertheless, empirical fragilities are limited to hurricane events due to the considerable number of damaged bridges and respective financial losses, motivating the interest of stakeholders [25,26]. Indeed, empirical studies concerning flood fragility curves focused on hurricane events [27–31]. To the authors best knowledge, flood fragility curves based on actual failures are not available for riverine bridges in mainland Europe. The need for empirical relationships is motivated through a learning process built upon evidence of damaged bridges. In addition, the results can be used to validate existing analytical models. Bridge owner records and post-disaster surveys improve the quality of available data, while a flood with hundreds of damaged structures ensures statistical quality [32]. The database of damaged bridges is thereby discretized in order to search for correlations between bridge features and the damage level. Nevertheless, reducing bridge collapse rates is a challenge for practitioners, as causes and mechanisms have specific features linked to each structure and its location [33]. Significantly, statistical techniques on a collapsed portfolio of bridges indicated that age, design enhancement and maintenance practices failed to reduce collapse rates against floods [34]. Therefore, further research is still needed on bridge failure mechanisms, including overlooked phenomena such as hydraulic force and driftwood clogging scenarios. The gap extends also to the fragility analysis, as flood received less attention in comparison to seismic hazards due to the complex dimension of implicated variables [35]. In this regard, FEMA P-58 seismic performance assessment methodology can be used to produce fragility curves for bridges subjected to floods [36]. The P-58's analytical formulation can be generalized, accounting for adjustments to hazards other than the seismic one. Metrics similar to those adopted in seismic fragility analysis are intended to be used, such as displacements of chosen structural elements [27,37]. However, the absence of such information for the majority of damaged structures, required the devising of original metrics. This approach is common in flood fragility analysis, as the literature indicates a variety of intensity measures—as opposed to seismic hazards—depending on available data and the investigated failure mode [38–40]. Recently, a seismic risk approach was used to assess structural damage caused by hydraulic force during the 2021 flood in a portfolio of buildings situated in the Ahr valley. The method employed a seismic damage classification as the basis for the flood damage model [41]. In the present study, evidence was collected from local authorities in the aftermath of the 2021 flood in Germany, classifying damage with the help of surveys, photos and bridge condition ratings. Then, a calibrated hydraulic model representative of the 2021 flood was employed to reconstruct the hazard magnitude at each bridge location [42]. Ultimately, the damage level is linked to a metric representative of the hydraulic force to produce fragility curves. Specifically, Section 2 illustrates the flood event; the bridge database, including recurrent failure modes; and the fragility generation. Section 3 is dedicated to the presentation and discussion of results. Section 4 illustrates the lessons learned and the limitations and briefly discusses

shortcomings in existing codes, while Section 5 is dedicated to the concluding remarks and future perspectives for this research.

## 2. Materials and Methods

The July 2021 flood mainly affected Germany, Belgium and Luxembourg, causing over 200 deaths, including 184 fatalities in Germany and 38 in Belgium [43]. The 2021 summer flood was reported to be the costliest flood in Europe and the deadliest in the last 30 years [44]. The flood caused extensive damage to infrastructure, especially bridges, leading to the isolation of many communities in rural areas [45]. Intermodal transport was also affected, having destroyed railway lines, for example, in the Ahr valley, where the reconstruction of the railway is expected to last several years [46]. Furthermore, the failure of the warning chain and the damage to critical infrastructure highlighted the need for improving disaster management and infrastructure planning [47].

The rainfall event according to Figure 1 was particularly severe in the Ahr river catchment region, as the basin received more than 60 mm/24 h, with two thirds of the area registering more than 100 mm/day [48]. In 2016, another flood occurred in the river Ahr, registering a discharge value close to that of a 100 year return period, but in terms of precipitation, the event was milder than the flood of 2021. What caused the water level to rise in 2016 was the duration of the rainfall, which started a week before the flood event, reducing the soil absorption properties [46]. In the 2021 event, most of the rain fell within 24 h, causing all secondary reaches to peak [49]. The consequent discharge accumulated in the main reach. In addition, the antecedent soil moisture condition was affected by the persisting depression in that area. Indeed, although the rainfall's peak occurred on 14 July, the event started on 12 July, continuing in the area throughout 12–13 July. On 13 July, the depression moved towards the Baltic Sea, before reverting to the affected areas on 14–15 July. Certainly, one of the causes of such an extreme precipitation could be found in the increased evaporation due to an exceptionally high sea temperature, compared to the average of that period [46]. Therefore, one can affirm that a climatic shift, or change, was part of the ingredients which led to this unprecedented disaster.

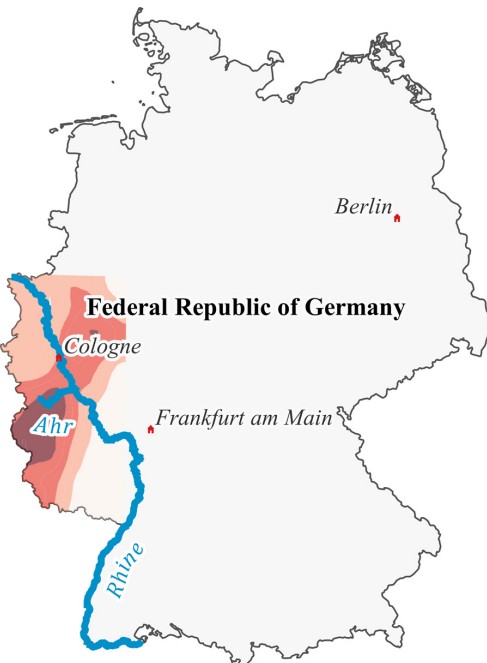

**Figure 1.** Daily accumulated precipitation (combined microwave–IR) 0.1 deg. (GPM GPM_3IMERGDF v06), 14–15 July 2021 [48]. The highlighted area represents the accumulated daily precipitation, with values ≥ 90 mm (darker region), 50 to 90 mm (intermediate), 30 to 50 mm (lighter region). Base map from NUTS250 [50], river shapefile from Waterbody-DE [51].

Before that event, the river gauge at Altenahr for a period of 100 years was estimated at 241 m$^3$/s and through a regression it can be assumed to have been 265 m$^3$/s for 200 years, with a R$^2$ = 0.99 (using discharge values for return periods of 2, 5, 10, 20, 25 and 50 years). However, as per Figure 2, the 2021 event's peak reconstructed by LfU was estimated at 991 m$^3$/s [52]. This discrepancy posed additional open questions regarding the treatment of flood events, including the suitability of extreme value distributions and the role of inline structures in producing backwater effects when debris accumulation occurs. Concerning the first question, the highest discharges that occurred in the Ahr river were estimated to be 1200 m$^3$/s at Dernau in 1804 and about 600 m$^3$/s at Altenahr in 1910 [53]. Interestingly, these events were not included in the flood risk assessment of the local authority, which is debatable from a risk management perspective, given that in 2021 similar values were registered, as per Figure 2 [6]. On the other hand, these rare events, if examined using the extreme value theory, would have led to return periods of about 10$^8$ years, highlighting a limit of these statistical models [54]. The other issue concerns anthropogenic reductions to the river's cross-section (e.g., parking lot in Altenahr) and bridges, causing increased water levels upstream [6]. Most notably, the interaction between bridges and drifting debris often caused clogging, resulting in a damming effect [55].

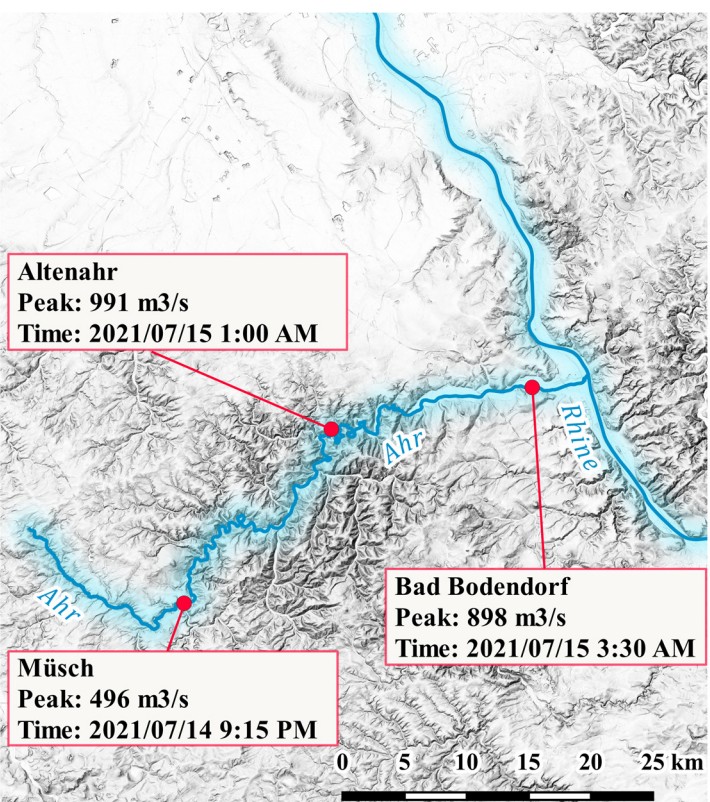

**Figure 2.** Spatial and temporal evolution of the peak discharge along the Ahr river according to the preliminary data from LfU [52].

This condition is visible in the upper part of the Ahr basin, where small streams still hold significant amounts of sediment and carry wood logs, as shown in Figure 3. This situation is not isolated, as in Germany, erosion rates easily exceed acceptable amounts [56]. Ultimately, the mobilization of driftwood is responsible for bridge clogging, increasing horizontal water thrust against decks and piers [57].

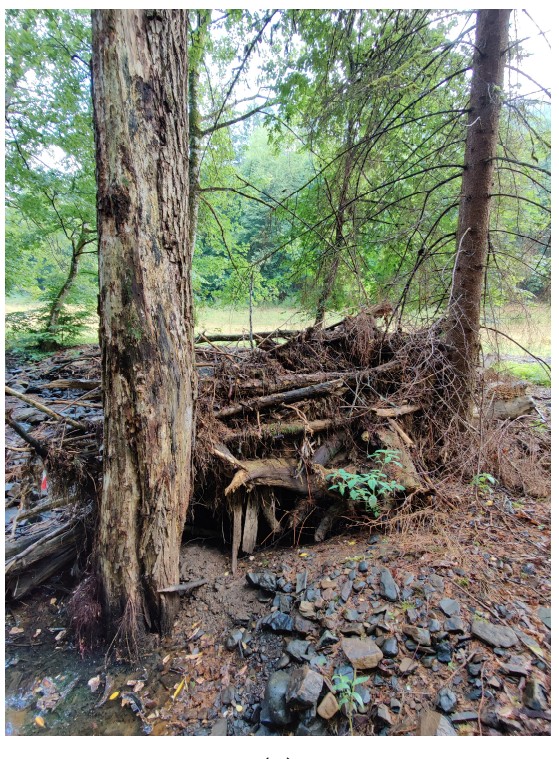
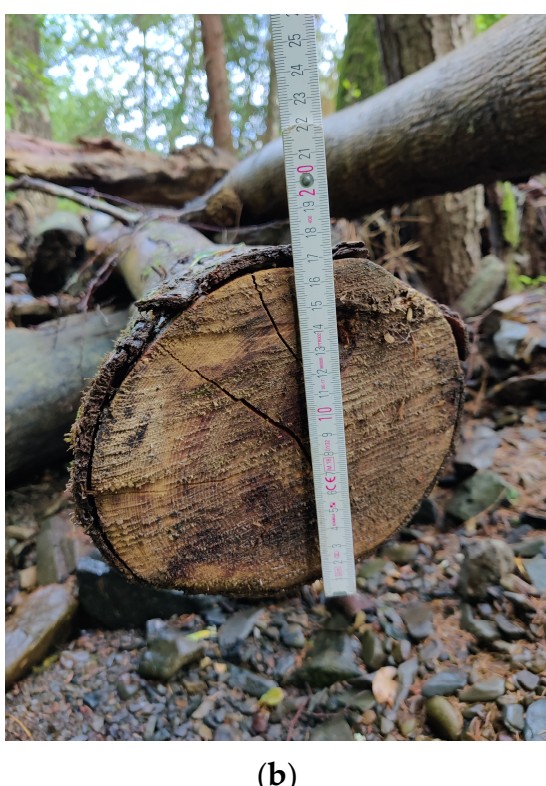

(**a**)                                                (**b**)

**Figure 3.** Debris damming in Eichenbach, tributary of the Ahr river: (**a**) lengths of carried wood logs of up to 15 m; (**b**) average diameter of carried logs (17 cm). Pictures taken by the first author.

### 2.1. Bridge Database

The affected bridges are located in two federal states: Rhineland-Palatinate (RLP) and North Rhine-Westphalia (NRW). According to the database of federal bridges in Germany, there are about 0.045 bridges over watercourses per square kilometer in NRW and 0.040 in RLP. This encompasses nearly 1500 bridges managed by the federal road authority in NRW and approximately 800 in RLP; however, it was not possible to determine the number of locally managed ones. The presented database comprised 250 bridges, including a vast majority of locally managed structures. In addition, culverts were not included in the database. Given the mentioned spatial pattern of rainfall, just 32% of the structures were located in NRW, while the remaining 67% were located in RLP. Of these, 99 bridges (60% of the total) spanned the Ahr river. Therefore, the present analysis concentrated on this watercourse. In Figure 4, the accumulated precipitation was overlaid with the kernel density of the 250 damaged bridges. Consequently, the spatial correlation between the magnitude of the weather event and the damaged bridges was highlighted.

The information on each asset was collected through a dedicated survey and integrated by incorporating additional data such as location, missing bridge features and cross-sectional elevation through 5 m and 1 m DEMs [58,59]. Then, a survey campaign took place to investigate details on failure mechanisms, collecting measurements and photographic material as per Figures 5 and 6. Specifically, two main types of failures were recognized: those due to scour and those caused by hydraulic force. In both cases, debris clogging exacerbated those phenomena.

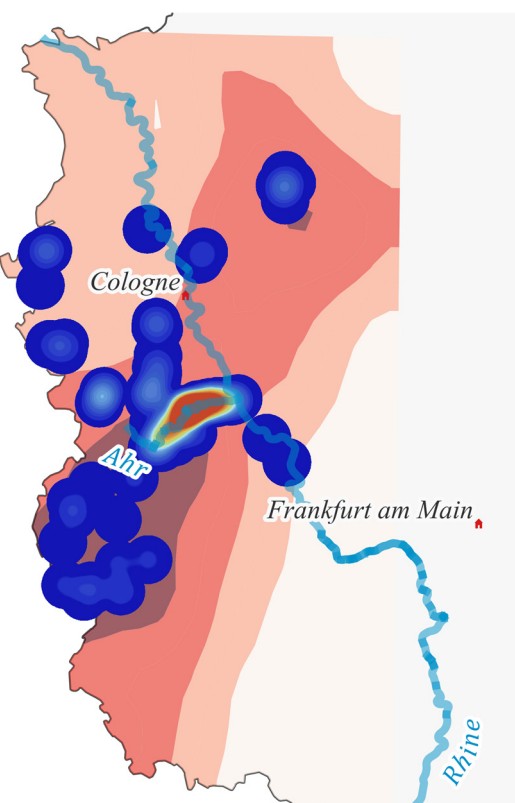

**Figure 4.** Spatial kernel density estimation of damaged bridges (blue to red dots). The Ahr valley was the most affected area. It can also be seen that the upper part of the river basin was affected by the greatest hydrologic loads (darker red on rain map: rainfall height ≥ 90 mm), inducing high hydraulic force along the stream. The other highlighted areas represent the accumulated daily precipitation, with values ranging between 50 to 90 mm (red), 30 to 50 mm (lighter red region). Daily accumulated precipitation (combined microwave–IR) 0.1 deg. (GPM GPM_3IMERGDF v06), 14–15 July 2021 [48]. Base map from NUTS250 [50], river shapefile from Waterbody-DE [51].

Scour is the erosion of soil from riverbed and riverbanks in the proximity of bridge foundations due to water flow. It is caused by the local hydraulic interaction between the structure and the streambed material. As scour depth increases, the lateral resistance of the soil supporting the structure diminishes, inducing foundation settling [3]. The survey campaign revealed the occurrence of different scour types in the Ahr river, including long-term riverbed degradation, local scour and contraction scour. Figure 5a shows a pit scour hole due to the 2021 flood event. The average depth of those pits is about 60 cm. A factor that contributes to the formation of these holes is soil erodibility, confirmed by the soil shrinkage at the bottom of the pit due to clay presence. These holes usually deepen and widen over time, becoming a significant concern for the bridge's structural safety. It is therefore essential to monitor the structures affected by that type of damage, preventing the scour from reaching its critical depth. This can be achieved via SHM (structural health monitoring), as innovations on the subject are increasingly applied to scour monitoring as well as during emergency management [60–62].

Figure 5b depicts the rightmost pier of the St. Nepomuk Bridge, also shown in Figure 5c, a masonry arch bridge built in the XVIII century. The bridge partially collapsed during the 2021 event due to scour in the approach fill, leaving the arch horizontal thrust unbalanced, causing its collapse. The bridge is also affected by scour on the instream piers, as seen in Figure 5b. As can be seen from this figure, the pier was built with a shallow foundation directly placed on riverbed stones. A scour depth of 70 cm was measured, although the erosion also affected the pier itself. Indeed, the aging mortar used for the bridge crushed easily under finger pressure. However, the overall scour condition of the

bridge should be investigated more broadly, as the riverbed on the upstream side of the bridge did not exhibit aggradation tendencies, suggesting long-term degradation. The situation is exacerbated by the flow contraction under the bridge, which locally increases the water velocity.

Figure 5d shows an example of channel flanking scour on a bridge with deep foundation. The water eroded the soil adjacent to the bridge abutment, exposing the pile heads. This type of scour widens the channel, increasing the risk of riverbank instability [63]. In the specific situation, the bridge did not collapse, but in many other situations along the Ahr river, channel flanking led to bridge collapse. To summarize, scour caused significant damage to masonry arch bridges, while reinforced concrete structures with deep foundations were slightly affected by it.

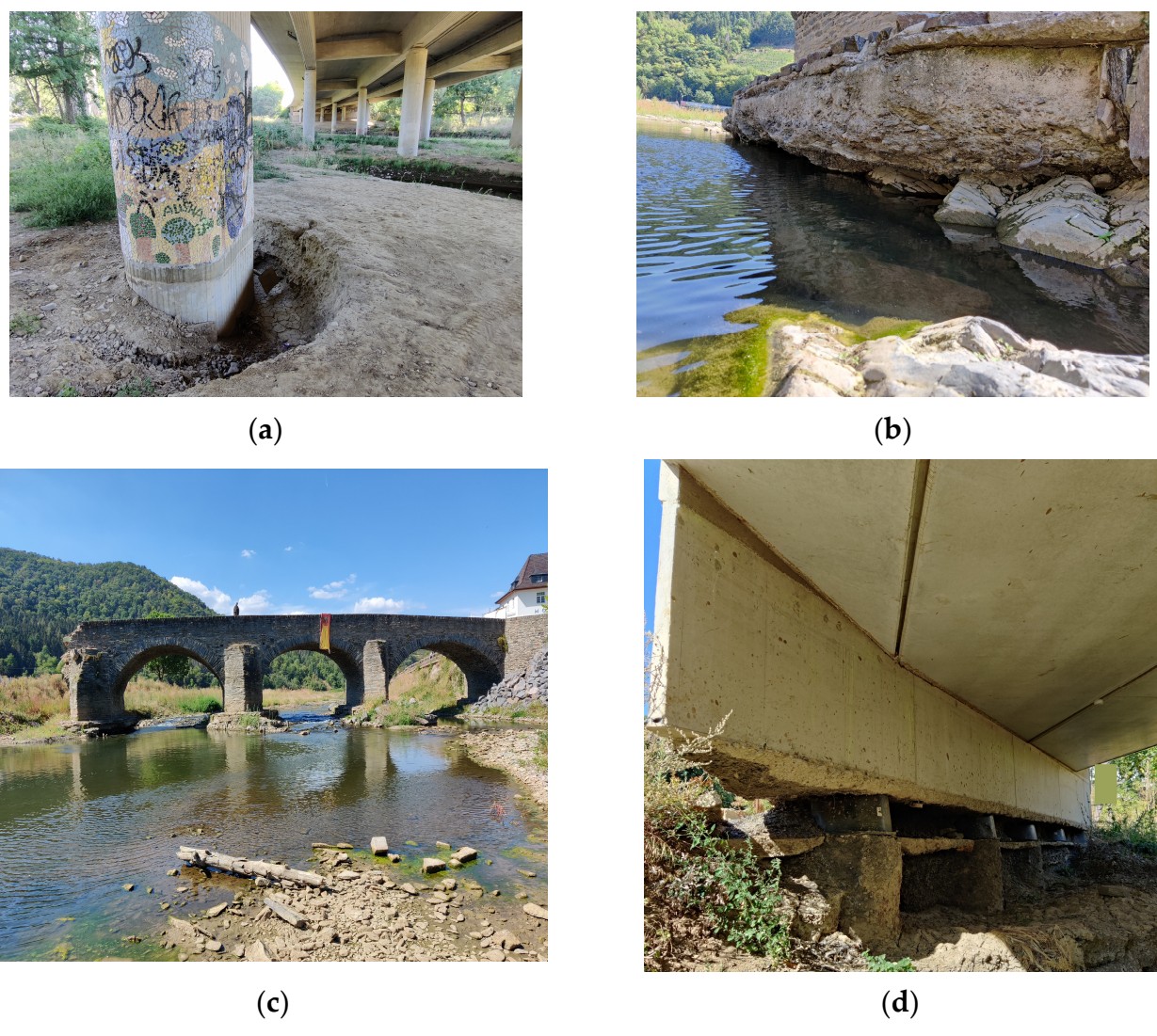

(**a**)　　　　　　　　　　　　　　　　(**b**)

(**c**)　　　　　　　　　　　　　　　　(**d**)

**Figure 5.** Surveyed scour mechanisms: (**a**) pit scour; (**b**) scour of pier; (**c**) scour of approach fill; (**d**) channel flanking scour. Pictures taken by the first author.

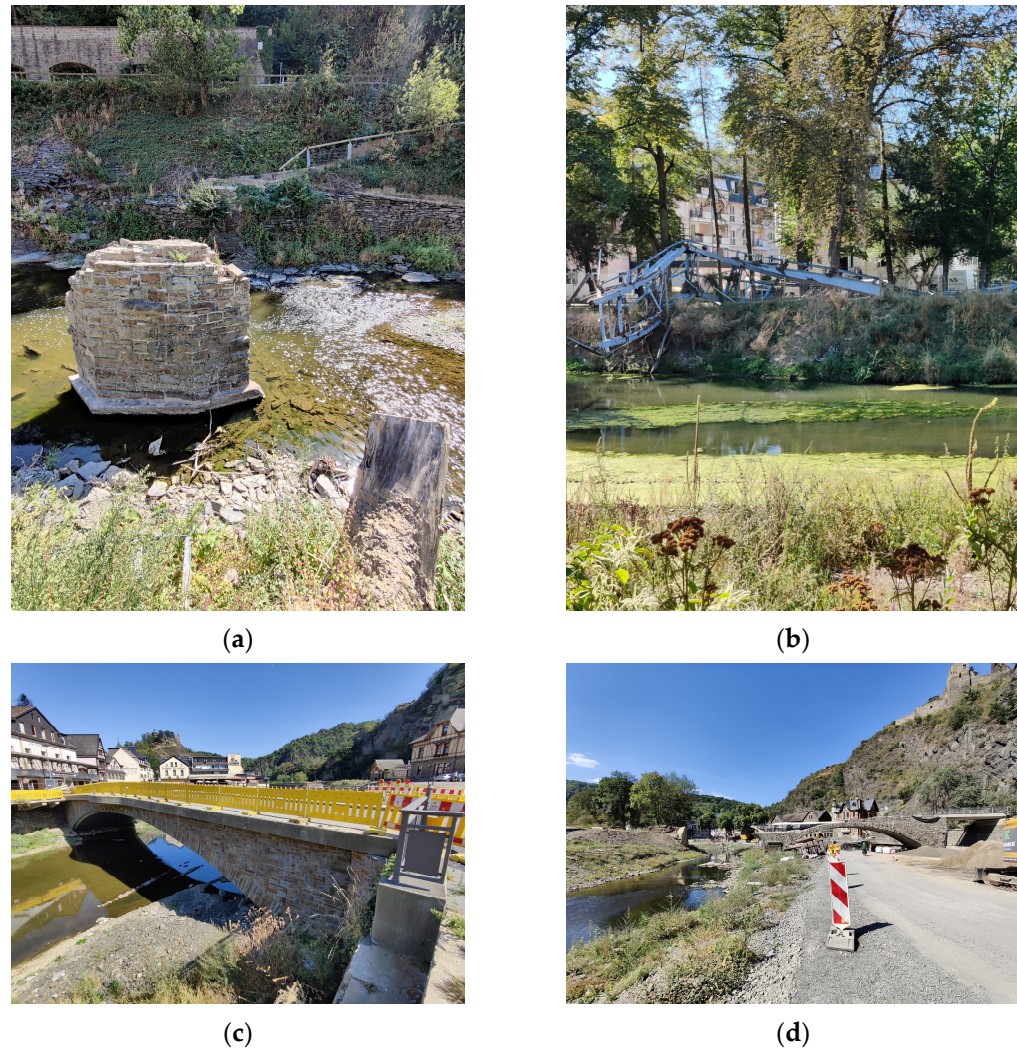

**Figure 6.** Surveyed overtopping mechanisms: (**a**) remaining pier of a dragged wooden deck; (**b**) girder of a dragged steel deck; (**c**) damaged railing due to overtopping; (**d**) combination of overtopping and scour, with the latter being responsible for triggering the failure. Pictures taken by the first author.

The other recurrent failures were triggered by deck overtopping, with a key role played by clogged debris. The size of the hydrodynamic forces was magnified by wood logs and carried material, resulting in damming effects with severe consequences for both the structure and its surroundings, inducing backwater effects.

Figure 6a shows the remaining instream pier of a wooden bridge, whose deck was found about 5 km downstream, while Figure 6b shows the steel deck of a footbridge dragged downstream a couple hundred meters from its original location.

From these two examples, one can observe that only lightweight decks suffered from dragging, but evidence in the aftermath of the flood showed that mixed steel–concrete road decks and a steel deck of a railway bridge also suffered from dragging [46]. During the 2021 flood, the hydraulic force against decks caused damage to all structures, with a clear trend: beam bridges with simply supported spans showed a higher vulnerability to dragging when compared to arch bridges. Indeed, no arch bridges that were overtopped experienced a full collapse. Severe damage was observed for masonry arch bridges, including the removal of infill material, carriageways and railings. Nevertheless, their failure was eventually only triggered by scour, as seen in Figure 6d. Overtopped arch bridges built in reinforced concrete sustained minor damage compared to masonry bridges, mainly to railings and parapets, as seen in Figure 6c.

Concerning the wood log impact and clogging, experimental campaigns have provided insights into the governing forces [64–66]; log jams at bridges significantly reduced structural safety, due to an increased flow impact area. By looking at the collected evidence, a key role in collapses was played by driftwood for both arch and beam bridges, while damage caused by uplift was seen only in wooden structures, given the higher buoyancy of the material and lack of evidence within the investigated structures.

### 2.2. Damage Categories and Bridge Condition Rating

The German Standard DIN 1076 regulates structural and traffic security of road infrastructures, with emphasis on the inspection and analysis of bridges, tunnels and culverts [67]. Each damage type is assessed and rated, justifying the reduction in structural safety, durability and/or traffic safety. In addition, guidelines support analysts in determining bridge condition ratings, with a grading system ranging from 1 to 4, including one decimal place [68]. The best condition possible for a structure is 1, while 4 is attributed to collapsed structures. The scale is not linear, thus for example, a structure rated 2.0 is not twice as safe as one rated 4.0. Under the same logic, the decimal point at threshold bounds should not be considered as a slight increment, i.e., when increasing a score from 3.4 to 3.5, the damage should be significantly different. The classification follows the scheme displayed in Table 1.

**Table 1.** Definitions of bridge condition ratings according to DIN 1076 [68].

| Rating | Structural Safety | Traffic Safety | Durability |
| --- | --- | --- | --- |
| 1.0–1.4 | Not compromised | Not compromised | Not compromised |
| 1.5–1.9 | Not compromised | Not compromised | Can be compromised in the long-term |
| 2.0–2.4 | Not compromised | Not compromised | Can be compromised in the medium-term |
| 2.5–2.9 | Not compromised | Can be compromised | Can be compromised |
| 3.0–3.4 | Is compromised | Is compromised | Extensively compromised |
| 3.5–4.0 | Extensively compromised | Extensively compromised | Extensively compromised |

In the database of bridges damaged in the 2021 flood, many structures were inspected by qualified surveyors in the aftermath of the event. However, for some structures, unfortunately those data were not available. For most of these bridges, other types of documentation was found, such as pictures and damage reports. To solve the issue of having quantitative information for one part of the database and qualitative information for the other part, expert opinion was employed to homogenize the two scales. A minority of structures did not have enough data to work with and were therefore not included.

As bridge ratings were semantically described, the process of attributing categories was facilitated, considering also the detailed description included in the DIN 1076, which was briefly recapped in Table 1. To balance granularity and accuracy, four damage categories were created:

1. undamaged—D1;
2. slightly damaged—D2;
3. moderately damaged—D3;
4. extensively damaged—D4.

The categories differ from HAZUS ones, which are slight, moderate, extensive and complete damage [27]. This discrepancy is mainly due to the different rating system associated with the structures. Indeed, the German bridge condition rating based on DIN1076 differs from that of the NBI (National Bridge Inventory) [68]. Therefore, the present categories were chosen according to HAZUS-based classification, which is employed in the existing literature on empirical fragility curves issued for bridges [28,29]. This choice was made in an attempt to facilitate further comparisons at research level but also maintaining a rigorous approach when following the damage levels reported in Table 1, which are described in the German standard DIN1076 [68]. Therefore, the only difference to the HAZUS classification system concerned structural collapse (German rating = 4.0), which

was associated with the complete damage reported in HAZUS but was included in the extensive damage category in DIN1076, given that the rating spans between 3.5 and 4.0, as per Table 1. Both in the USA and Germany, ratings are given by qualified experts. From this perspective, the condition rating is based on experts' judgment on the safety domain boundary, which is linked to failure mechanisms. In addition, even though ratings of 1.0 to 2.9 refer to a safe, non-compromised structure, it was decided to create two categories, distinguishing them based on the damage that could cause durability issues in the structural integrity. For the other classes (ratings above 2.9), the distinction presented in Table 1 was maintained. The qualitative scheme is presented in Table 2.

**Table 2.** Equivalence of bridge condition ratings to a qualitative damage level, based on the structural safety parameters.

| D1 | D2 | D3 | D4 |
| :---: | :---: | :---: | :---: |
| 1.0–1.9 | 2.0–2.9 | 3.0–3.4 | 3.5–4.0 |

### 2.3. Statistics of Population

Information about the construction date for 184 bridges out of 250 was obtained. According to the age distribution and the number of failures per each structural typology, beams and arches were the most affected types, with a total of 125 and 82 bridges, respectively. Interestingly, there was a high number of newly built beam bridges, mainly footbridges. From the analysis of the post-disaster evidence, many of these structures collapsed due to unexpected water activity, often in combination with driftwood blockage, which increased the horizontal thrust on the decks, as seen in Figure 6a,b.

Then, the bridge inspection records antecedent to the flood were analyzed, comparing those in the present database to all bridges which had been federally managed. Structures that were present in both databases were deleted from the federal database. From this analysis, a statistically significant worse average condition rating was found among damaged arch bridges compared to their counterparts in the federal database. There were no similar differences for the other bridge typologies. The data also allowed us to make the same comparison by filtering the federal database to analyze only bridges over rivers in the states of NRW and RLP. To this end, the Kruskal–Wallis (K-W) test was performed on the two databases. A K-W test was used as an equivalent to ANOVA but for non-parametric data. The factor was the bridge typology, accounting for the following categories: a) beam and box girder bridges and b) arch bridges. Before the K-W test, the data were tested against Levene's assumption, resulting in a rejection of the null hypothesis with a confidence level at $\alpha$ 0.05 [69]. A K-W test was performed, resulting in significance at a $p$-value of 0.0001. To shed light on individual subgroups, a nonparametric post hoc test was used, with the $p$-value corrected according to Bonferroni's assumption [70].

The results showed the greatest difference in condition ratings among beam bridges, with an adjusted $p$-value of 0.0010, while among arch bridges it was 0.0043. On average, the condition rating of the damaged bridges measured before the flood was statistically worse compared to that of the undamaged population. These differences obviously increase if the rating assigned to the damaged bridges after the flood is used. The test was significant as reported, but not all the undamaged structures were subjected to the same hazard magnitude. To better explain this point, a correlation between the damage level and an intensity measure representative of the hydraulic force on bridges was searched for. The triggering mechanisms were selected based on evidence from surveys and damage reports. Then, the predominance of one mechanism over the other (scour over hydraulic force) was found and was highlighted in both Figures 7 and 8, with respect to deck typology (beam vs. arch bridges) and weight (masonry and concrete decks, i.e., heavy, vs. steel and wooden ones, i.e., lightweight). Then, the FEMA P-58 method was used to draw fragility curves starting from these correlations [36].

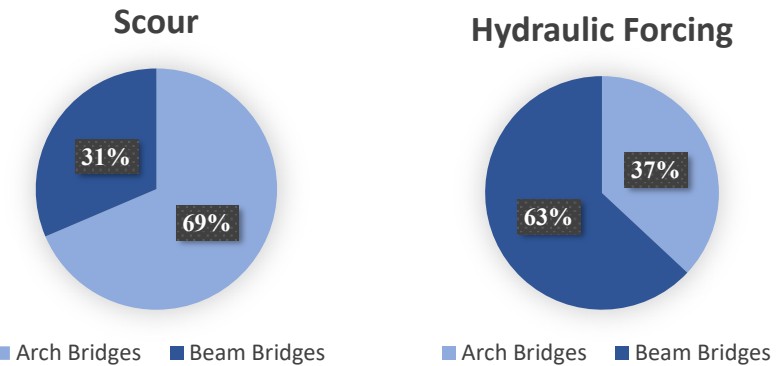

**Figure 7.** Percentages of arch and beam bridges with respect to the two damage mechanisms.

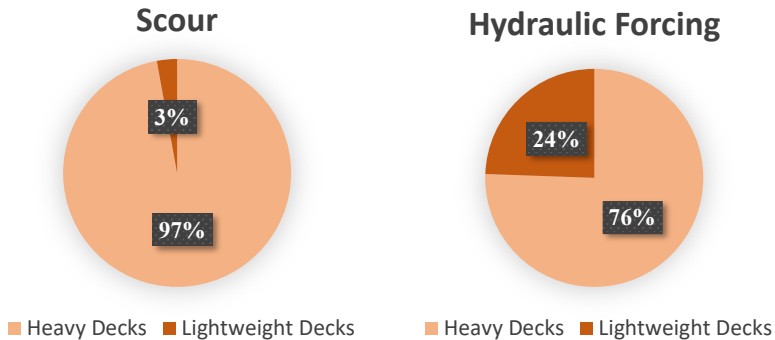

**Figure 8.** Percentages of heavy and lightweight bridge decks with respect to the two damage mechanisms.

### 2.4. Fragility Curves Generation

Fragility curves link the hazard intensity to the damage experienced by structures. Various metrics can be chosen to represent the hazard. In case of floods, it is common to use flow discharge or water elevation, depending on the situation. In the present case, a metric called $h^{**}$ was used, which is the ratio between the flood height and the bridge deck elevation, as per Figure 9. The symbol $h^{**}$ was chosen to differentiate it from $h^*$, called the 'inundation ratio', defined in flume experiment study as $h^* = (h_u - h_b)/s$, where $s$ is the thickness of the bridge deck.

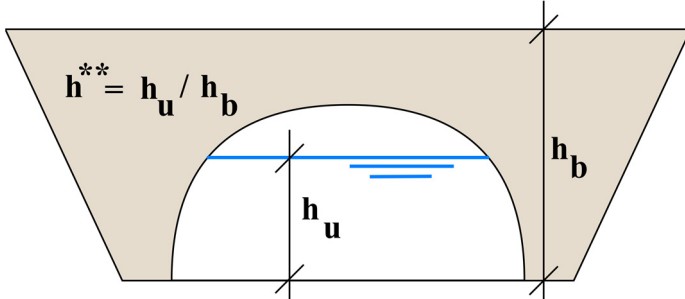

**Figure 9.** Definition of $h^{**}$ as fragility intensity measure.

The metric allowed us to use a single category for various bridge geometries, as the relative height had a good correlation with the recorded damage levels.

In other flood phenomena, such as hurricanes, the storm surge can be a good alternative, but in regions such as those of the case under examination, varying bridge clearances pose an issue. Bridges in mountainous environments usually have low clearances, while downstream bridges have higher clearances. Despite this difference, both upstream and downstream bridges sustained the same damage levels. However, in such cases, recorded water elevation and flow discharge were different. Conversely, $h^{**}$ was comparable; in this

way, from a structural point of view, $h^{**}$ helped to homogenize the population from the hazard point of view.

The other considered intensity measure was water velocity, but additional information to distinguish between different soils was not available, resulting in considerable uncertainties. In addition, the simulated velocity had to be considered upstream of the bridge, as local contractions could significantly increase it.

The reconstructed peak discharge, as per Figure 2, was useful to investigate many aspects of the flood process. In such a context, Apel, Vorogushyn and Merz developed an hydraulic model of the Ahr river between Altenahr and Sinzig, studying the effect of houses on the increased volume of water [71]. As mentioned, the water discharge was similar to the 1804s, although only minor damage was observed at the time. The study by Apel, Vorogushyn and Merz is of particular interest for infrastructure managers, as the increased water height directly affected bridges in the sense that buildings subtracted areas that would have otherwise been occupied by water, as in 1804. This effect increased the water levels, supporting the use of water height to characterize the hazard intensity. Although discharge represented a better intensity measure, the required data to obtain that information were affected by high uncertainty, as the river overtopped bridges causing a pressurized flow underneath many structures.

The fragility curves are obtained by means of the following expression:

$$F_d(r) = \Phi\left(\frac{\ln\left(\frac{r_i}{\mu_d}\right)}{\beta_d}\right) \tag{1}$$

where the parameters of the distribution are obtained by using the maximum likelihood estimation method:

$$\mu_d = \exp\left(\frac{1}{n_d} \cdot \sum_{i=1}^{n_i} ln(r_i)\right) \tag{2}$$

$$\beta_d = \sqrt{\frac{1}{n_d - 1}\sum_{i=1}^{n_i}\left[\ln\left(\frac{r_i}{\mu_d}\right)\right]^2} \tag{3}$$

Here, $F_d(r)$ is the fragility estimated at intensity $r$ for damage state $d$. The parameters $\mu$ and $\beta$ are the mean and standard deviation values of the lognormal cumulative distribution, respectively, and $n$ is the number of elements or specimens of empirical data. The subscript $d$ is used to differentiate between damage levels. Two tests were employed to validate the model: the goodness of fit test and a criterion to manage outliers. With the first test, it was checked whether the data actually followed the hypothesized normal behavior. The goodness of fit to a normal distribution is usually ensured through Kolmogorov–Smirnov's (K-S) test. However, in this case, the fit was imposed through the calculation of $\mu$ and $\beta$. Thus, K-S tables were no longer valid. To this end, Lilliefors' test was used, which employs a modified K-S tables [72]. The management of outliers was carried out using Peirce's criterion [73]. Confidence intervals were computed by adopting the uncertainty provided by the digital elevation model (DEM). According to the DEM data, the reported error is equal to $\pm 0.3$ up to 1 m, depending on the terrain type [58] and assuming that the measurements are being carried out with an accuracy of 0.5 m. Considering the 95% confidence intervals for the data points, Figure 10 was obtained.

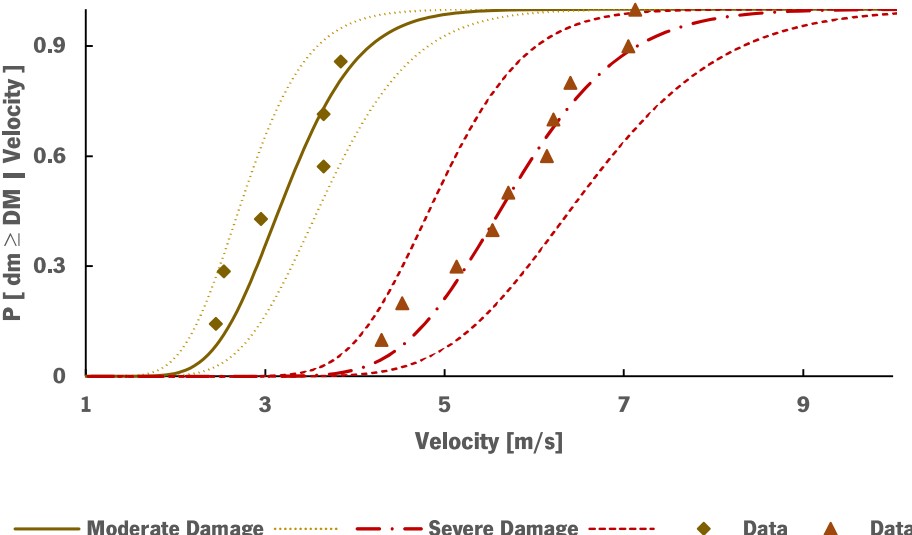

**Figure 10.** Confidence intervals for the moderate and severe damage states. Velocity data considering only arch bridges. The 95% confidence intervals are highlighted together with the mean value of the fitted distribution.

Nevertheless, since the flow velocity was obtained from the hydraulic model for only a portion of the Ahr river (below Altenahr town), the fragility curves displayed in the next section are only presented with $h^{**}$ as an intensity measure. Concerning this point, the water elevation was reconstructed in the upper part of the Ahr basin (above Altenahr), using markings and topographical measurements via a 1 m LiDAR map with an accuracy of 0.3 m; while in the lower part of the basin, the same procedure was double-checked against the calibrated hydraulic model [42], as mentioned in Section 1.

## 3. Results

The data were clustered in two different ways: based on the deck building material and based on the typology. For the first cluster, lightweight structures, such as those made of steel and wood, were separated from heavier structures, typically built with concrete or masonry. Wood and steel decks were considered together as evidence indicating similar failure modes (i.e., deck dragging and uplift). Then, by using the same principle, concrete and masonry bridges were also considered together. Therefore, the term "lightweight" (LWY) structures was used to indicate steel and wooden decks, and "heavy" (HVY) structures represented those built with masonry and concrete. The nomenclature was kept in Figures 11 and 12. Concerning the failure mechanisms, scour typologies were grouped into a single mechanism, in accordance with Figure 5, with the purpose of separating those from the mechanisms caused by hydrodynamic dragging and uplift (i.e., hydraulic force), as seen in Figure 6.

The chart in Figure 11 shows the probability severe damage in both lightweight and heavy decks for the two failure modes. Scoured bridges exhibited higher probabilities of being damaged in both lightweight and heavy decks, compared to the hydraulic force failure mode. Nevertheless, the behavior was similar for both damage mechanisms when the deck was not overtopped ($h^{**} < 1$). This confirmed that such damage was not influenced by the deck material but by other factors associated with different failure mechanisms, such as the scouring of instream piles. For $h^{**} > 1$, there were differences among lightweight and heavy decks under scour, but these were still minor when uncertainties were included (i.e., 5% and 95% confidence intervals) and are not represented in Figure 11 for the sake of clarity but are shown together with the velocity as IM in Figure 10.

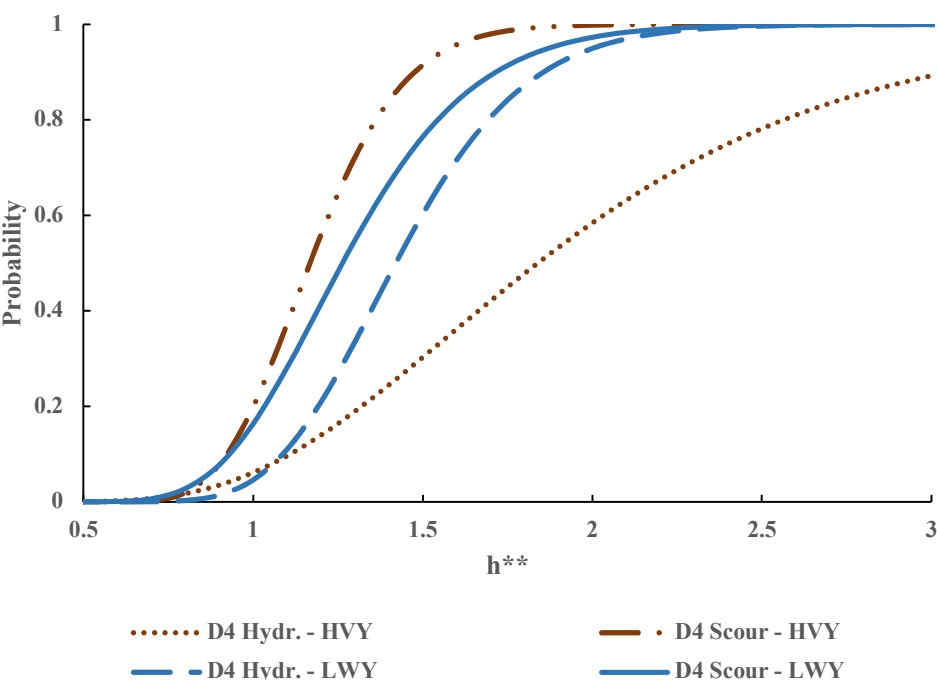

**Figure 11.** Fragility curves for severely damaged heavy (HVY) and lightweight (LWY) bridge decks under scour and hydraulic force (Hydr.) scenarios.

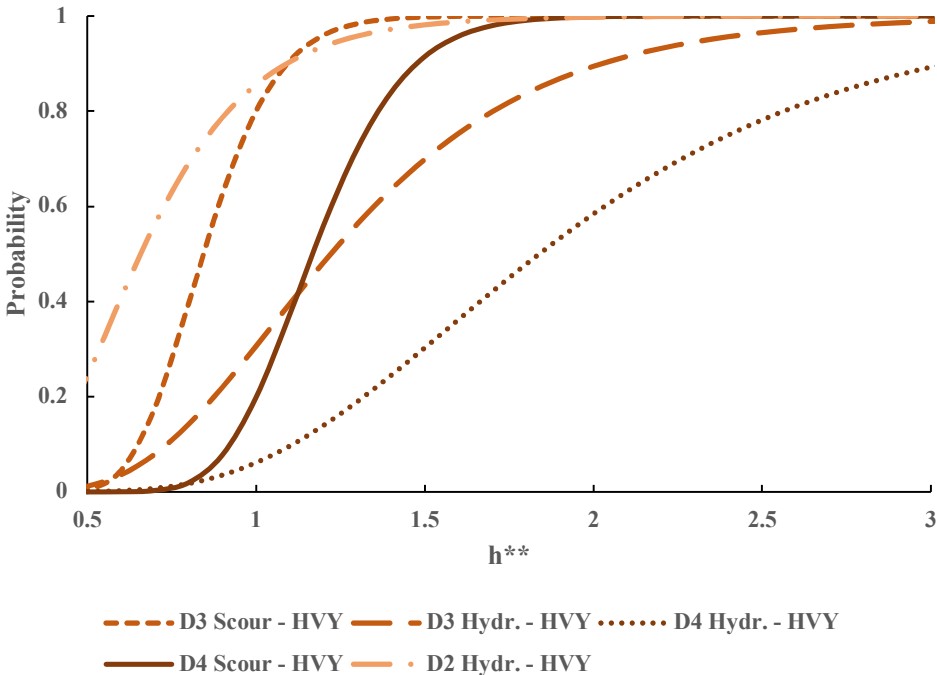

**Figure 12.** Fragility curves for heavy bridges (HVY) under scour and hydraulic force (Hydr.) scenarios accounting for all the damage levels.

Indeed, when the confidence bands (5–95%) are included, the biggest difference between scour and hydrodynamic mechanisms was observed for heavy decks in the overtopping interval ($h^{**} > 1$). Thus, heavy decks had less probability of being severely damaged at high water stages compared to when scour occurred. The same also happened for lightweight decks, but the curves overlap when considering the confidence bands. Differences also existed when damage caused by high water stages were considered. As

expected, lightweight decks exhibited greater probabilities of being severely damaged compared to the heavier ones.

When slight and moderate damage levels were considered, there were only heavy decks, as seen in Figure 12. Hence, lightweight structures exhibited only severe damage, as they were less robust to hydraulic force. Consequently, Figure 12 shows sequential damage states for heavy decks only, subjected to the same failure mechanisms. As expected, the behavior for heavy decks under scour was more severe than that under the hydraulic force damage mechanism. At low water stages, slight damage was more likely to occur. The damage level was also a function of debris carried by the flow, but the lack of data did not allow us to assess the impact of this factor on the fragility model.

Another result concerned the lack of slight damage under scour events. This confirmed that erosion is a moderate to severe problem for heavy decks. An important aspect is that when these bridges are overtopped ($h^{**} = 1$), there is an 80% probability of observing a moderate to severe damage in case of scour, while the probability reduces to 36% in cases of water thrust, as a damage mechanism is triggered.

Considering damage levels, the probability that scour caused a moderate damage is 33% (hydrodynamic loads is therefore 67%), while for a severe damage the probability rises to 78%, leaving hydrodynamic loads with a probability of causing the remaining 22% of occurrences.

For the second cluster, beam structures, including trusses and box-girders, were separated from arch bridges. Figures 13 and 14 present the results for beam and arch bridges, respectively. By looking at Figure 13, it is shown that among beam bridges, moderate damage is missing. Then, combining the fragilities for bridge material and typology, it was identified that the moderate damage for heavy decks in Figure 12 only occurs in arch bridges. A common feature among the results is the severity of the scour mechanism, which led to higher damage probabilities for a given $h^{**}$.

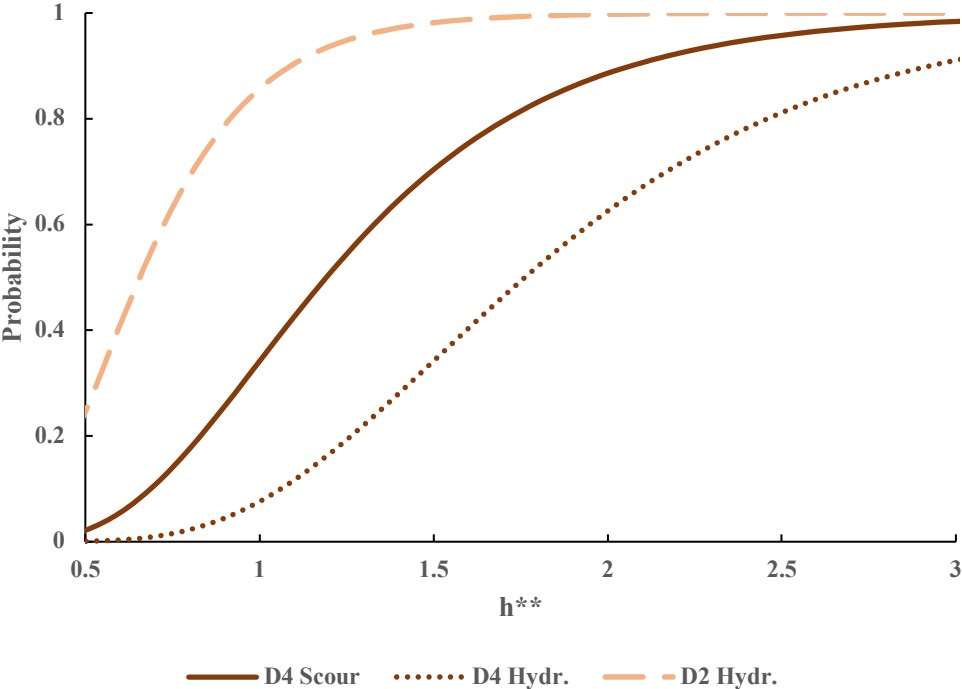

**Figure 13.** Fragility curves for beam bridges under scour and hydraulic force (Hydr.) mechanisms.

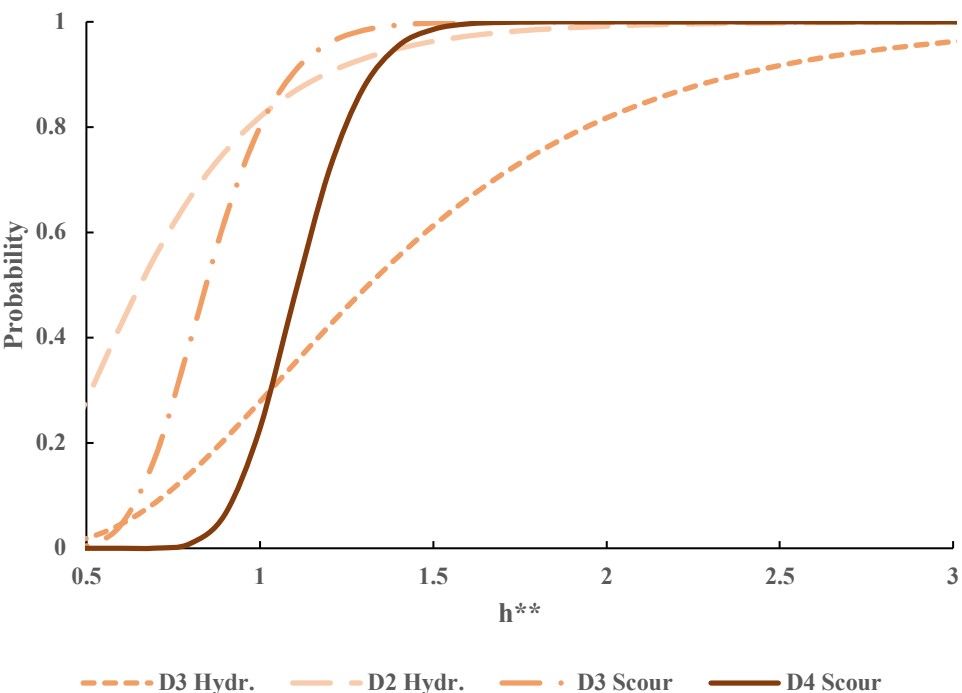

**Figure 14.** Fragility curves for arch bridges under scour and hydraulic force (Hydr.) scenarios.

Another observation concerns the absence of arch failures due to hydraulic force mechanism, as per Figure 14. Then, the severe damage caused by hydraulic force in Figure 12 is attributable to beam bridges. This opens up a major issue on whether arch bridges do experience collapse due to water thrust, and which hydraulic force component (drag or uplift) causes the most damage. While for the dragging action there is agreement among studies and data, concerning the uplift component, the studies by Falconer et al. [74] and Majtan, Cunningham and Rogers [75] are in disagreement with those of Jempson [76], Kerenyi et al., [77], Oudenbroek et al., [57] and Dean [78]. Indeed, Falconer et al. and Majtan, Cunningham and Rogers suggested a positive uplift mechanism for the arch, called upthrust, while flume experiments on hydrodynamic forces on bridge decks from the other authors reported negative uplift coefficients for most of the submergence ratios encountered in floods. However, the dynamic effect is mitigated by the Archimedes' thrust, which may eventually become predominant at higher submergence ratios, because of less negative dynamic uplift at higher water stages. The collected evidence from the bridges damaged in the 2021 flood in Germany are compatible with a positive drag and a negative uplift, as none of the arch bridges exhibited the failure mechanism described in Falconer et al. but instead suffered damage due to the combined effect of wood clogging and drag force. In this regard, the main problem is the high blockage of an arch against the flow, resulting in considerable drag, contributing to the removal of backfill and leaving the arch itself standing against the flow [79]. In addition, the impact of hydraulic force against the intrados often causes slight damage, i.e., masonry and mortar detachments, exposed rebars and parapets, among others. This phenomenon is shown in Figure 14, where slight damage occurred in partially submerged arch bridges, $h^{**} < 0.5$.

On the other hand, drag forces on beam decks are milder than those on arch bridges, due to the lower blockage. However, beam bridges exhibited severe damage due to water thrust, especially in simply supported decks, as seen in Figure 6a,b.

A superstructure's weight optimization can lead to failures in cases when high water is expected, as also demonstrated in the literature [27,29]. Lastly, slight damage (D2) had almost identical probabilities for both typologies (see Figures 13 and 14). This can be explained through empirical evidence, as arch and beam bridges experience different phenomena, which however, can be ranked under the same damage category. To summarize, within the studied database, arch bridges are more robust than beam bridges in high water.

Nevertheless, beam bridges tend to suffer less damage from hydraulic force, due to their lower blockages against the flow.

Regarding scour, it was observed that arch bridges suffered moderate and severe damage, while beam decks only experienced the latter. However, in terms of more severe damage, scour in arch bridges is more serious than in beam decks, as the damage probability rapidly increases once the bridge is overtopped.

One can therefore conclude that arch bridges are more prone to scour and debris clogging, although their structural behavior is more robust than beam bridges, which is confirmed by the presence of a moderate damage level.

## 4. Discussion

Damage reports used in this research were retrieved from local authorities, bridge condition inspections and integrated surveys in the aftermath of the 2021 flood in Germany. The intensity measure, called $h^{**}$, was chosen based on available data as the ratio between the water stage upstream bridges and the deck elevation. Nevertheless, existing literature demonstrated the relevance of geomorphologic indicators on the bridge collapse probability, suggesting that the failure mechanism can be significatively influenced by the location and hydraulic conditions of the stream [28]. In the present work, the aggregated geomorphologic indicator used in Germany to rank rivers was tested for usage [52], but no correlation was found with the selected intensity measure. This can be attributed to the aggregation level of sub-indicators in the aforementioned metric. Therefore, the explained variance was too low to proceed further. Concerning the hydraulic model, the flood event was reconstructed by Apel et al., [42] for the Ahr river (Germany) by using a 2D model calibrated on the hydrograph from LfU [52]. The damage levels were chosen based on the semantic description of the DIN 1076 bridge condition ratings [68] and the ranking method used in empirical fragility models [27,28]. However, for the collapse event (DIN1076 rating = 4.0), the damage was classified as extensive, as opposed to HAZUS, where a collapse is represented as a complete damage [27]. This discrepancy was highlighted in Section 2.2 and is due to the classification used in the DIN1076, as within the 3.5–4.0 interval, structural, traffic and durability safety are ranked as extensively compromised. The generation of fragility curves was developed according to the FEMA P-58 method and was adapted to floods [36]. To assess the goodness of fit and manage the outliers, we employed Lilliefors' test and Peirce's criterion, respectively. The failures were grouped by the triggering failure mechanisms; either scour- or hydrodynamic-related force. Although the influence of clogged debris has been pointed out, it was not possible to estimate this factor due to a lack of data. Then, bridges were clustered by observing trends between failure mechanisms and the deck material, separating lightweight structures from heavier ones. Then, bridges were also categorized based on their structural typology, distinguishing beam decks from arches. The results suggested that beam bridges subjected to water overtopping experienced a higher probability of failure compared to arches, although another internal subdivision among beam bridges had to be made. Indeed, lightweight beam decks exhibited even higher vulnerability to hydraulic force compared to heavier ones. However, beam bridges tended to have a lower occurrence of damage type than arches, due to their lower blockage to the flow. Nevertheless, the real failures demonstrated that arch bridges are not likely to collapse under high hydraulic force, often reporting slight to moderate damage. It should also be pointed out that there is a disagreement among studies concerning the magnitude of hydraulic uplift on arch bridges. In order to shed light on this point, a shift towards a probabilistic approach to account for hydrodynamic actions on decks is encouraged. To this end, Pucci et al. [80] presented a novel methodology to compute fragility curves caused by hydraulic force and driftwood actions for varying discharges.

When scour was considered the triggering mechanism, beam bridges usually collapsed, while arches reported a more robust behavior, showing moderate damage. However, beam bridges experienced lower scour-induced damage rates compared to arches.

Concerning existing codes, currently, the Eurocode 1 includes a specific limit state for horizontal water thrust on decks but only during bridge construction [81]. For the in-service bridge portfolio, the safety margin is represented by a given clearance on top of the 100- or 200-year flood level. On the other hand, standards such as the AS5100:2017 account for these failure mechanisms and provide practitioners with design charts to confirm the magnitude of hydrodynamic coefficients [82]. Indeed, the evidence collected in the aftermath of the 2021 flood on bridges confirmed the relevance of hydrodynamic actions during high water, stressing the need to provide practitioners with reliable tools to evaluate such failure modes during the construction of new and the assessment of existing bridges.

## 5. Conclusions

Fragility curves represent an important step in the financial risk assessment of existing bridge stock. This paper addressed this issue by developing fragility curves based on actual failure data. This analysis suggests that the cause for the high number of collapses is multifaceted. On one hand, climatic changes are increasing both the frequency and magnitude of extreme events, leading to unforeseen actions on structures. On the other hand, unexpected forces—such as overtopping—could represent a serious hazard to bridge stock. In addition, certain standards, such as the Eurocode 1, deal with hydrodynamic thrust on bridge decks only during the bridge construction and account only for the dragging limit. The Australian AS5100:2017 instead offers a holistic methodology to be applied throughout the structure's life, including drag, uplift and overturning limit states. Overall, this work has demonstrated through evidence collected after the 2021 flood in Germany that the current deterministic approach is not able to consider the high uncertainties related to the climate change, and therefore, we strengthen the call for a shift towards a probabilistic—or semi-probabilistic—approach for the computation of hydraulic forcing on bridges.

**Author Contributions:** Conceptualization, A.P., D.E. and R.H.; methodology, A.P.; software, A.P. and D.E.; validation, A.P., L.G. and H.S.S.; formal analysis, A.P. and L.G.; investigation, A.P., D.E. and R.H.; resources, J.C.M. and R.H.; data curation, A.P. and D.E.; writing—original draft preparation, A.P.; writing—review and editing, H.S.S. and L.G.; visualization, L.G. and J.C.M.; supervision, R.H. and J.C.M.; project administration, H.S.S.; funding acquisition, A.P., H.S.S. and J.C.M. All authors have read and agreed to the published version of the manuscript.

**Funding:** The first, fourth and fifth authors acknowledge that this work was partly financed by FCT/MCTES through national funds (PIDDAC) under the R&D Unit Institute for Sustainability and Innovation in Structural Engineering (ISISE), under reference UIDB/04029/2020, and under the Associate Laboratory Advanced Production and Intelligent Systems ARISE, under reference LA/P/0112/2020. This work was supported by the FCT Foundation for Science and Technology under Grant SFRH/BD/145478/2019.

**Institutional Review Board Statement:** Not applicable.

**Informed Consent Statement:** Not applicable.

**Data Availability Statement:** The data presented in this study are available on request from BASt, Bundesanstalt für Straßenwesen.

**Acknowledgments:** The authors would like to thank Heiko Apel for providing the results of the calibrated hydraulic model for the river Ahr.

**Conflicts of Interest:** The authors declare no conflict of interest.

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
