# Peer review of "Fragility Analysis Based on Damaged Bridges during the 2021 Flood in Germany"

_applsci, doi:10.3390/app131810454_

Round 1

Reviewer 1 Report

This paper investigates the fragility curves based on the failure of bridges in  2021 flood in Germany. The topic is very interesting and practical. The manuscript is suggested to be revised in the following aspects before acception.

1. On line 13, please type the full name of CAT since it's first used in paper.

2. On line 173-176, the author's contribution should not be descriped here.

3. On line 312, what does information mean, please explain.

Minor editing of English language required

Reviewer 2 Report

In this context, empirical fragility curves were generated by referring to actual failures occurred in the 2021 flood in Germany. To the same aim, a calibrated hydraulic model of the event was used. Data was collected through surveys, damage reports and condition ratings from bridge owners. This study sheds light on existing vulnerabilities of bridges to river floods, discussing specific areas in which literature data are contradictory. The paper also strengthens the call for a shift towards a probabilistic approach to estimate hydraulic forcing in bridge design and assessment. In general, the authors make efforts to present their work. A few questions are requested in the following comments to be duly handled.

1. In this paper, the authors make the fragility research based on collapsed bridges, which is a good work. The reviewer suggests to focus more on the short comings of the existing references. Meanwhile, the reviewer suggests to add some literature review of fragility approaches, see 10.3390/math11061294; 10.1080/13632469.2021.1887011; 10.1016/j.jobe.2022.105716.

2.A brief flowchart is suggested to be given to present the main procedure in the analysis.

3. In the fragility analysis, the limit states and damage states are important aspect. How do the authors consider this aspect?

4. Please add the horizontal and vertical axes in the figure (e.g., figure 8-12)

5. The conclusion is suggested to be further refined (e.g., novelty of the paper, limitations, or further investigations).

Quality of English Language can be further checked.

Reviewer 3 Report

The article

"Fragility analysis based on collapsed bridges during the 2021 flood in Germany"

By Pucci et al,

delves into the significant impact of floods as one of the most financially burdensome natural disasters, particularly when it comes to their role in causing, as estimated, over half of all bridge collapses. Indeed, also in accordance with this Reviewer’s own experience, hydraulic causes are the most prominent first cause of bridge collapses.

The objective of the presented study is to construct empirical fragility curves based on actual bridge failures that occurred during the 2021 flood in Germany. The methodology presented by the Authors involved collecting data from surveys, damage reports, and condition ratings provided by bridge owners, resulting in a comprehensive database encompassing 250 bridges. Furthermore, the research underscores the importance of considering not only the probability of damage but also the likelihood of a specific triggering mechanism occurring when evaluating bridge vulnerabilities during floods.

The paper is well written and the results are explained in detail, with overall good and fundamental considerations.

Nevertheless, before full acceptance can be granted, the following points need to be assessed by the Authors.

 1.      The Authors present a very interesting (and potentially very useful) set of analytically-defined, empirical fragility curves. This is most probably the most important contribution of this work to the existing knowledge. However, it would be better to detail much more the context. At page 9, it is only (too succinctly) stated that ‘The categories were chosen according to [the] existing literature on empirical fragility curves issued for bridges [25,26].’

2.      Regarding the Bridge Database: if this Reviewer understood correctly, the Authors state that, during a single event, 250 bridges (out of 1500) collapsed. According to this Reviewer’s experience, both the absolute number and the percentage over the total seem to be extremely high. Perhaps it depends on what the Authors labelled as a ‘bridge’ (any span length?) or as a ‘collapse’ (any partial and/or non-fatal failure?). In any case, this should be discussed in more detail.

3.      The analysis uncovered two predominant categories of failure mechanisms during flood events: those induced by scour (the erosion of the riverbed around bridge piers) and those triggered by hydraulic forcing (pressure exerted by the flowing water). The predominancy of these two failure mechanisms should be discussed in further detail; which is the one more common for the several structural typologies investigated in the database?

4.      Related to the previous remark, significantly, the study found that the extent of damage inflicted on bridges during floods is heavily reliant on the bridge's typology, followed by the weight of its deck. This implies that certain bridge designs demonstrate greater resilience against the stresses imposed by floodwaters. However, again, this should also be quantitatively, and statistically, evaluated.

5.      Also, perhaps the last two points may be biased by the specific structural typologies most commonly used in the relatively small region of interest, and/or its geographical and hydrogeological characteristics?

6.      Importantly, the paper draws attention to areas within the existing literature where contradictory data on bridge vulnerabilities to river floods may be present, emphasizing the need for clarification and standardization in this field. It would be of real interest, however, to read the Authors’ own recommendations in this regard, based on the experience accumulated during this study.

7.      Similar to the previous point, the authors of this study advocate for a shift towards a probabilistic approach in estimating hydraulic forces during bridge design and assessment, as this approach promises a more comprehensive understanding of the risks associated with river floods. It would be interesting to have further details in this regard, also considering applications to other countries and/or regions outside of Germany.

8.      Discussing bridge collapses and scouring, it would be worth mentioning the large body of scientific literature on this subject, such as https://doi.org/10.1007/s13349-020-00398-0, https://doi.org/10.1002/stc.3028, https://doi.org/10.1016/j.istruc.2020.12.073, and https://doi.org/10.1080/15732479.2022.2048030

9.      The Conclusions are a bit too lengthy, and perhaps they can be shortened or split into two sections: a longer “Discussion” and a brief “Conclusions”.

The Quality of English is overall above average and only requires some minor corrections.

Round 2

Reviewer 2 Report

The authors have responded to my comments, and now it can be accepted. 

Reviewer 3 Report

This Reviewer is fully satisfied with the current version of the manuscript

The Quality of English Language is good.